# Built Environment Factors Influencing Prevalence of Hypertension at Community Level in China: The Case of Wuhan

**Hongjie Xie** [1] , **Qiankun Wang** [1] **, Xilin Zhou** [1],* , **Yiping Yang** [2] **, Yuwei Mao** [1] **and Xu Zhang** [3]

1   School of Civil Engineering and Architecture, Wuhan University of Technology, Wuhan 430070, China;
    h.j.xie@whut.edu.cn (H.X.); wangqk@whut.edu.cn (Q.W.); sashiharaaa@whut.edu.cn (Y.M.)
2   Wuhan Branch of Chinese Center for Disease Control and Prevention, Wuhan 430010, China;
    jaqwjwjkk@163.com
3   School of Resource and Environment, Wuhan University of Technology, Wuhan 430070, China;
    zhangxu1212@whut.edu.cn
*   Correspondence: zhou.xilin@whut.edu.cn; Tel.: +86-18986174097

**Abstract:** This paper studies the correlation between built environment factors and the prevalence of hypertension in Wuhan, a typical city in central China. Data were obtained from a regional epidemiological database, which is the 2015 Epidemiological Survey of people under 65 years in 144 communities. The prevalence of hypertension was analyzed in five components based on the WHO framework (land use, transport, accessibility, green space, and socioeconomic status). Results indicated built environment factors have significant correlations with the prevalence of hypertension ($p < 0.01$). The road network density, gymnasium cost, income, medical facilities cost, walkability index, and land use mix (LUM) were statistically significant. Other indicators did not pass the significance test. The spatial models fit better than the multivariate linear model.

**Keywords:** healthy city planning; prevalence of hypertension; built environment factors; empirical research





## 1. Introduction

With the advancement of medical technologies and economic development, the field of population health has experienced an epidemiological transition from infectious diseases to noncommunicable diseases (NCDs) [1]. The main NCDs, such as cardiovascular disease, diabetes, cancers, and chronic respiratory disease, are the leading cause of death and are a large burden globally. NCDs are often described as "lifestyle" diseases because the main risk factors are overconsumption of food, physical inactivity, smoking, and alcohol abuse. In recent decades there has been increasing recognition that the urban environment and built environment can have a significant (although complex and difficult to quantify) impact on human health. There have been only a few studies that have attempted to examine how the built environment impacts on the health of residents. Evidence supporting the association between the built environment and NCDs is far from convincing and remains ambiguous and fragmented [2,3]. The problem with most research is that case studies often tell stories with complex details or prove positive claims, rather than derive inductive hypotheses or test useful ideas that build on well-established theories [4]. Few studies have tried to establish a holistic conceptual framework or explain the mechanism of the built environment impacting health.

There has been relatively little work on NCDs and almost nothing of relevance on the relationship between built environment and NCDs in China [5]. Most of the Chinese scholars follow the direction of their colleagues in the United States and Europe. However, the built environment in China is very different from that of North America and

Europe. Whether the healthy city theories or approaches originating from the West are still applicable to China, or not, needs to be verified [6].

That the built environment affects health and well-being has been confirmed by many studies [7,8]. The Lalonde report issued by the Canadian government in 1974 first confirmed that the human habitat is one of the main factors affecting health [9]. It was the first recognition that the built environment has a considerable impact on physical activity [10–12]. Since then, the association between the built environment and NCDs has attracted increasing attention from decision-makers and researchers. The eligible studies published have found there are three main vectors of built environment and health outcomes: one is *physical activity* [13], the other is the *urban foodscape* [14], and the third is the *restorative environment* [15].

In the WHO/Europe's *Evidence Review of Spatial Determinants of Health* [16], the main urban components that determine health are land use patterns, transport, green space, and urban design. In other recent holistic framework studies on the built environment and chronic disease, Nieuwenhuijsen and Khreis et al. proposed that land use, facility accessibility, mobility, physical activity, environmental exposure, and social participation are important components of a healthy city [17,18] and examine the eligible studies between 2005 and 2015 worldwide to assess the influence of built environmental attributes on cardiovascular disease (CVD) risks. It was found that neighborhood environmental attributes were significantly associated with CVD risk. Residential density, traffic, recreation facilities, street connectivity, and a highly walkable environment were associated with physical activity. Highly walkable environments, fast food restaurants, and supermarket/grocery stores were associated with blood pressure fluctuations [19]. Sarkar et al. found that in areas with a low residential density (1800 p/km$^2$) or less, a positive association between density and obesity was observed. If density was more than 1800 p/km$^2$, it was negatively associated with obesity, in contrast [20]. In addition, it was reported that green space is beneficial for mental health [21,22], not only designated green space types such as parks, but also, and in general, street greenery [23].

Several studies have been conducted in various parts of the world to understand the health impacts of the built environment [24,25]. So far, empirical studies on the causal correlation between built environment factors and NCDs such as hypertension have rarely been seen. This is partly due to the fact that health outcomes are an evolutionary process of long-term exposure to risk factors, including confounding physiological and psychological factors, and even social behaviors; most studies only point out the relationships [26]. Even Reid Ewing, one of the leading scholars in the field of the healthy city, cannot assert unequivocally that urban sprawl causes obesity but can only state that the two factors are significantly related [27]. It is widely accepted that NCDs are caused by comprehensive risk factors and it is difficult to establish a reliable model of their causal association [28,29]. Various etiological mechanisms exist in the complex causal chain of NCDs. For example, many risk factors cause the same disease, but on the other hand one factor may be related to multiple chronic diseases. Bird et al. found no linear association among the built environment factors, physical activity, and health outcomes [30].

The earliest consensus about the relationship between the built environment and NCDs is the impact of the built environment on physical activity; for example, whether the design of streets is conducive to walking and cycling, whether there is a mix of land uses that encourages walking and cycling, and whether there are green spaces and sports fields for outdoor activities. There has been a large body of work on this [27,31,32]. Frank et al. (2004) found that every 25% increase in the land use mix (LUM) reduced the overweight and obesity rate by 12.2% [33]. Li et al. (2008) observed that for every unit increase in the LUM (entropy index) in the Portland metropolis, the rate of overweight and obesity in the elderly decreased by 25%. Liu and Yang (2016) compared four communities in Dalian and found that diversified land use and high street connectivity promoted an increase in physical activities in the elderly [34]. However, contrasting results were also reported. Cerin et al. (2007) observed that the association between LUM and walking was not significant in

research on the community environment and destinations [35]. Forsyth A. et al. found that the LUM was negatively correlated with an increase in physical activity [36]. Ewing and Cervero (2010) reported that the assessment methods provided inaccurate results in a meta-analysis on LUM and physical activity. The entropy algorithm of LUM may not be the best tool for determining the association between land use and physical activities [37]. Green spaces and landscapes also encourage physical and psychological health [38], expediting the recovery of patients [39].

This study follows the WHO's framework and selects land use, transport, green space, and urban design as the four components of the built environment. *LUM* and *FAR (Floor area ratio)* were selected for *land use*; density of road network and destination accessibility for *transport*; normalized difference vegetation index (NDVI) for *green space*; and walkability index for *urban design*, respectively. In addition, socioeconomic status is also an important determinant of health. Thus, demographic and socioeconomic factors were added to the research framework. The theoretical analysis model is shown in Figure 1.

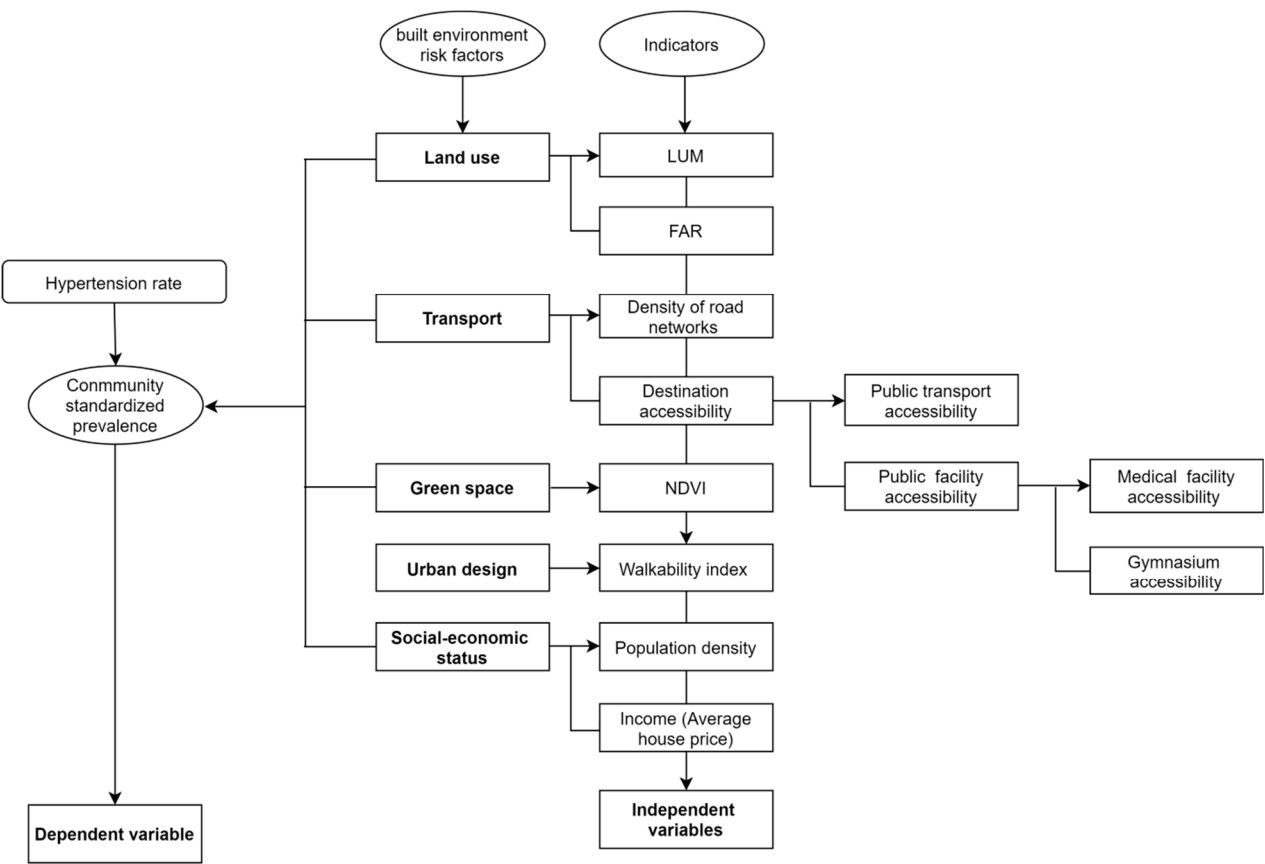

**Figure 1.** Multiple regression model of the prevalence of hypertension and the influencing factors of the built environment.

## 2. Study Area

The study district is one of the seven main districts of Wuhan city, located on the north bank of the Changjiang River. It has a wide variety of land types, including the downtown area with high urbanization and building density, a developing area, as well as some agricultural land and forest. Wuhan is the economic and geographical center of central China, so it is a good representative Chinese city, with an area of 70.25 km$^2$, 137 communities, and 7 villages (Figure 2).

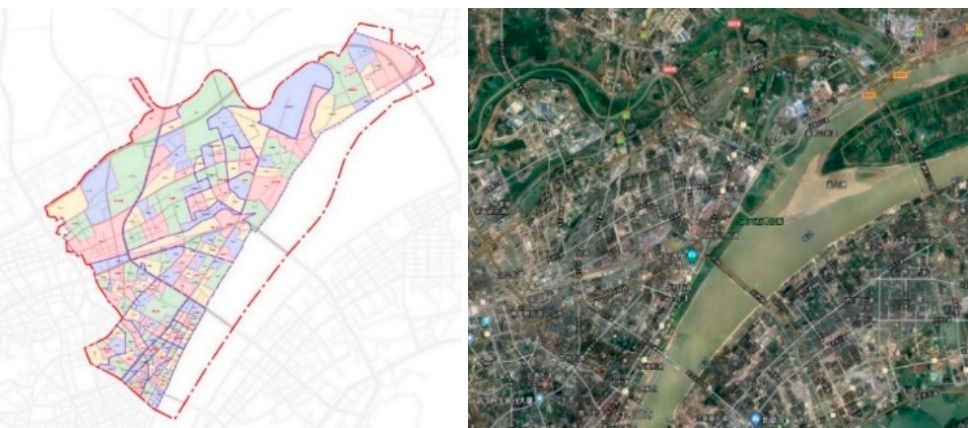

**Figure 2.** Research scope and community grid.

## 3. Materials and Methods

### 3.1. Collecting Hypertension Data for Visualized Images at Community Level

In this cross-sectional analysis, prevalence data were collected from an epidemiological survey, which was collected from 2012 to 2015 in Jiang'an district in Wuhan. It was collected using a multi-stage stratified and random sampling method from 2012 to 2015. So far it is one of the most reliable hypertension epidemiological datasets [40]. After removing duplicates and sorting out the original data, a total of 40,113 adult hypertensive patients aged 18–99 (86.7% between 40 and 69) remained in the dataset. The number of male patients (22,411) was higher than that of female patients (17,702).

In this study, the data of people over 65 years were excluded to minimize the influence of age. According to the *China Cardiovascular Diseases Report 2018*, the prevalence rate of hypertension between 17 and 64 years is 23.2%, whereas it is 56.0% in people over 65 years old, affecting more than half of the population that age and losing statistical significance [41].

The raw data were positioned by a coordinate conversion system using geocoding software and linked to a community base map (Figure 3). The prevalence data of hypertension per 10,000 people were calculated in each community (Figure 4). Arcgis software was utilized to see if there was spatial aggregation in the prevalence of hypertension at community level with visualized images.

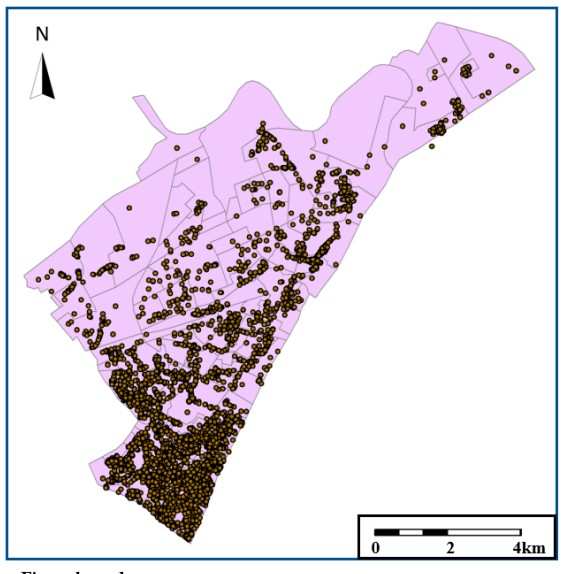

**Figure legend**

● Hypertensive patient

**Figure 3.** Distribution of hypertension patients in the Jiang'an district in Wuhan.

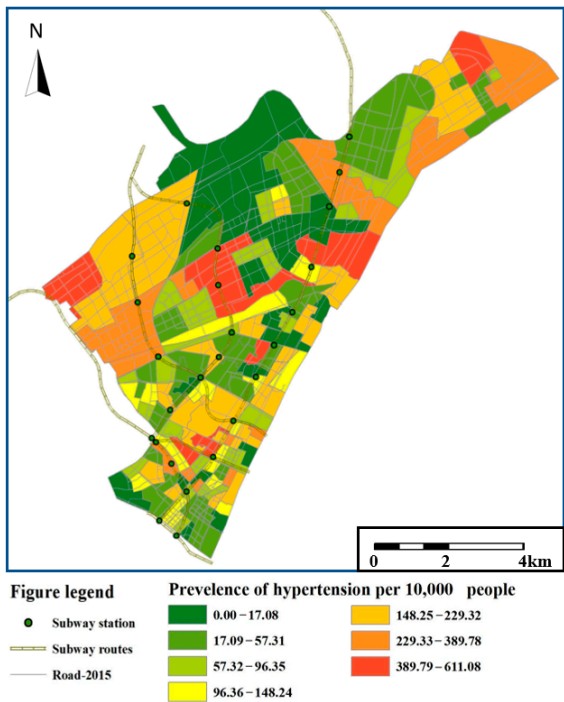

**Figure 4.** The prevalence data of hypertension per 10,000 people at community scale.

### *3.2. Calculation of Built Environment Indicators*

### 3.2.1. Land Use

For the land-use components, we chose LUM and floor area ratio (FAR) as indicators.

- Land use mix (LUM):

LUM is used to evaluate the degree of mixing of various land uses in a certain area. It was difficult to create a quantitative calculation method for LUM before Cervero and Kockelman first proposed the nine-square grid model [42]. At present, the entropy model proposed by Frank et al. is widely recognized by scholars [43] (Figure 5).

$$LUM = -\sum_{i=0}^{n} Pi * \ln P_i / \ln n \tag{1}$$

where *n* is the number of land uses, and *Pi* is the percentage of the area attributed to land use *i*. The four land uses of residential, commercial, office, and institutional were utilized to calculate LUM.

- Floor area ratio (FAR):

Due to the small research scale and most buildings being high-rise buildings in the study area, *FAR* was selected as the land-use intensity indicator. The *FAR* is the ratio of the total building area to the land area (2) (Figure 6).

$$FAR = \frac{\sum_{i=1}^{n} S_i * n}{S_C} \tag{2}$$

where $S_i$ is the building area of a single building, *n* is the number of buildings in the block, and $S_C$ is the block land area.

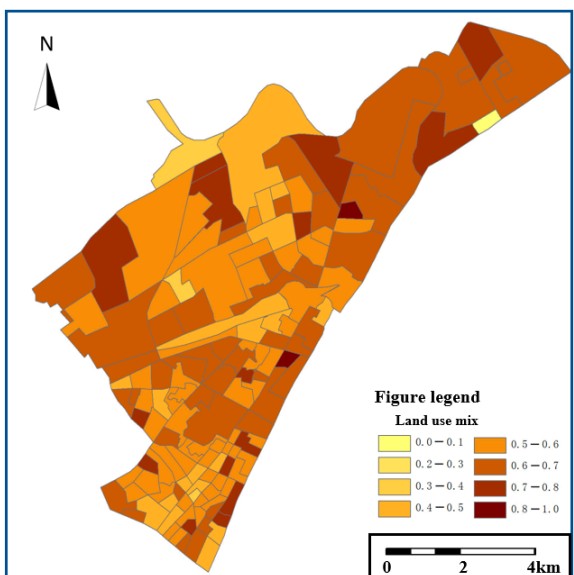

**Figure 5.** Land use mix (*LUM*) of the communities in the Jiang'an district.

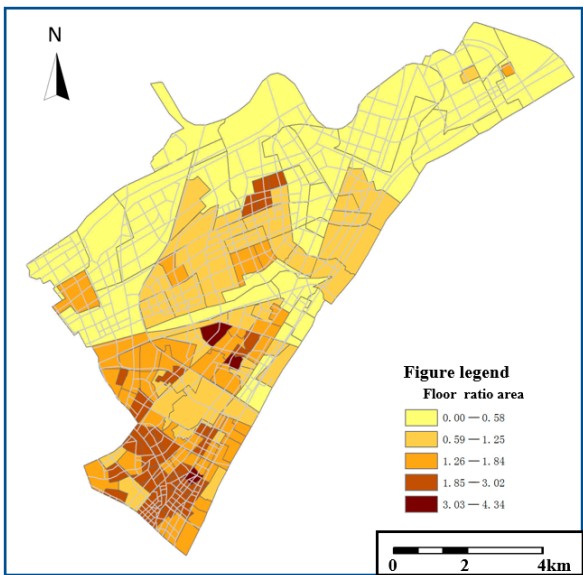

**Figure 6.** Floor area ratio (*FAR*) of the communities in the Jiang'an district.

### 3.2.2. Transport

The density of the road network *(RoadDen)* and the public facilities' accessibility were selected to characterize transport.

- Density of road networks:

The density of road networks (*RoadDen*), which is the ratio of the total length of all the roads to a unit area, can be used to characterize the connectivity level. A high *RoadDen* indicates better walkability in the neighborhood and promotes more physical activity, such as walking and cycling (Figure 7).

- Destination accessibility:

Destination accessibility is a variable used in the 3D models proposed by Ewing and Cervero (2001) and represents the ability to travel between locations [44]. The calculation of accessibility is relatively complicated and various. In this study, we used public transport and public facility accessibility to measure destination accessibility.

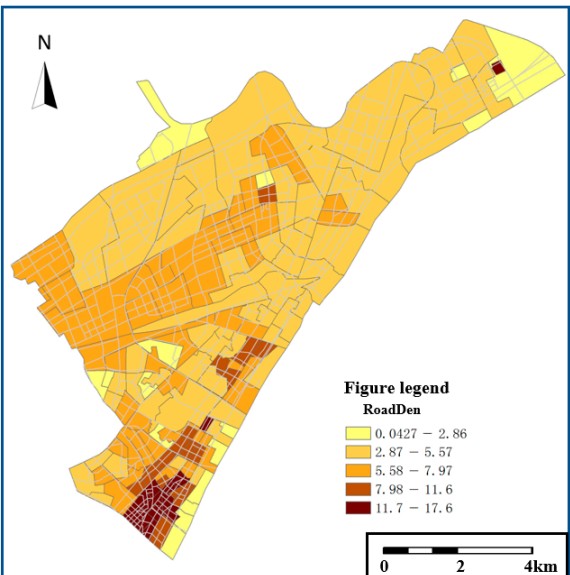

**Figure 7.** Density of the road network in the communities in the Jiang'an district.

- Public transport accessibility:

The density and coverage of bus stations were selected as two indicators to calculate bus accessibility *(BusIndex)*. The density of bus stations is the number of bus stations per unit area (the weight of subway stations was twice that of bus stations) (Figure 8). The coverage of stations is the percentage of the area served by the bus and subway stations. To identify the service radius of public stations, we checked the literature. Referring to some of the existing studies [45,46], we identified that the bus stations covered a 300 m range and the subway stations covered an 800 m range. The weights of bus station density and coverage were set to 0.4 and 0.6 respectively (3) (Figure 9).

$$F = \frac{\varphi}{S_n} \times 0 \cdot 4 + \frac{S_i}{Sn} \times 0.6 \tag{3}$$

where $\varphi$ is the number of bus stations (subway stations*2), *Si* is the area covered by the bus station, and *Sn* is the area of the community.

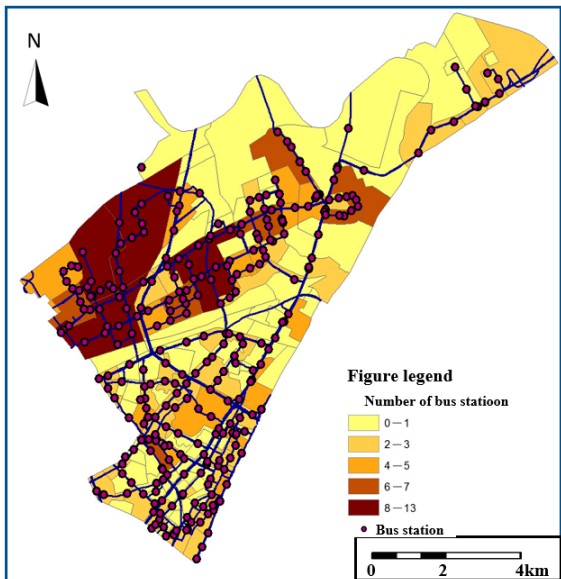

**Figure 8.** Number of bus stations in communities in the Jiang'an district.

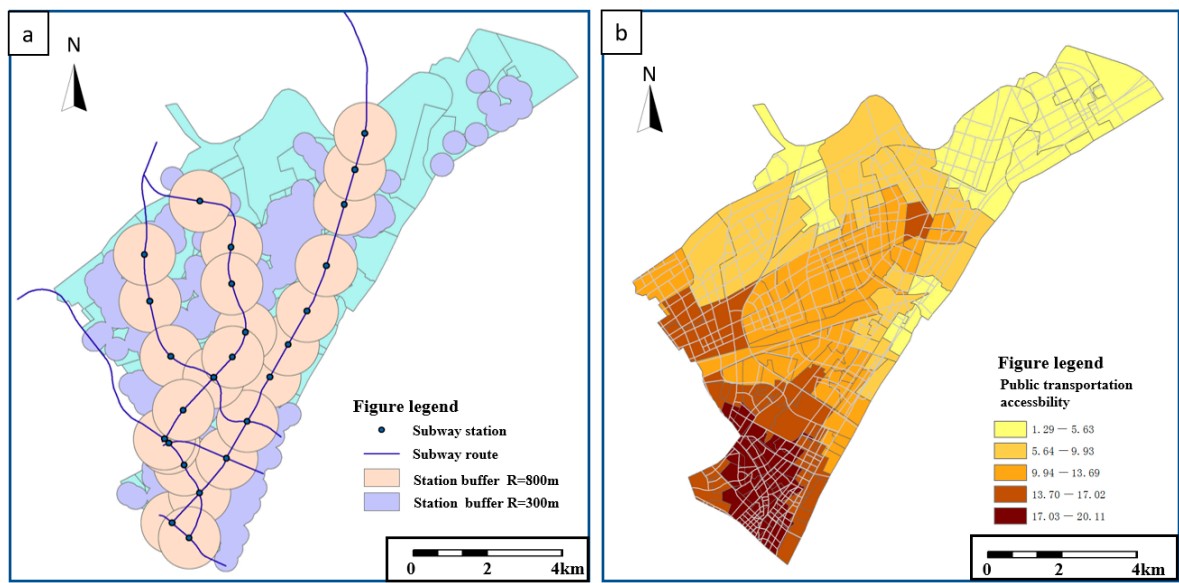

**Figure 9.** (**a**) Coverage area of bus and subway stations; (**b**) public transport accessibility of the communities.

- Accessibility of public facilities:

The distance cost to Class IIIA hospitals and gymnasiums, which are closely related to health, were selected as the indicator of the accessibility of public facilities. The distance cost is the shortest distance from each community to the nearest Class IIIA hospital *(MedCost)* and gymnasium *(GymCost)* based on street connectivity (Figure 10).

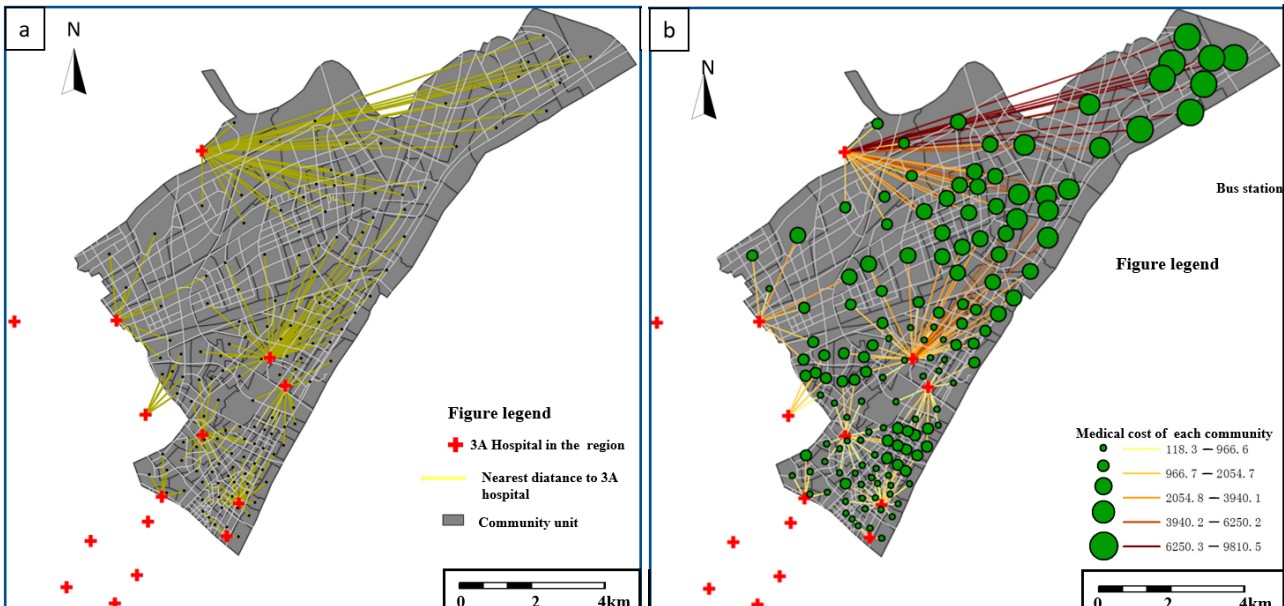

**Figure 10.** (**a**) Distance from each community to the nearest medical facility; (**b**) analysis of the distance cost of medical facilities in the communities in the Jiang'an district.

### 3.2.3. Green Space

The green area ratio is often used as an indicator of green space. Due to rapid changes in Chinese cities, the planning maps often lags behind actual developments. Therefore, we used another indicator, the *normalized difference vegetation index (NDVI)* obtained from remote sensing data, to characterize the actual distribution of green space in the study area.

### 3.2.4. Urban Design

Morphological parameters, such as sky view factor (*SVF*), street width to building height ratio (*W/H*), building coverage ratio (*BCR*), and building surface fraction (BSF) etc., are key factors of urban design. While morphological parameters were linked with the vibrancy of the city, they were too many to be examined in this study. Besides, the emotional and subjective feelings of residents on urban design make it difficult to obtain a quantitative measure [47]. The *walkability index (WI)* is a recently emerged indicator that evaluates the vibrancy of cities [48], indirectly reflecting the quality of urban design [49,50]. Many algorithms exist to calculate the WI. This study used the walk score algorithm, which is relatively mature and is recognized by most scholars worldwide [51]. The walk score was calculated as the usage of various public facilities with different weights based on the travel behaviors of pedestrians. The walking distance attenuation, intersection density, and road length were considered to improve the accuracy of the *WI*, Equation (4) (Figure 11).

$$S = \sum_{i=1}^{n} w_i \times n_i \times P_i \tag{4}$$

where $w_i$ is the weight of the evaluation factor $Xi$, $n_i$ is the number of $X_i$, and $P_i$ is the distance attenuation coefficient.

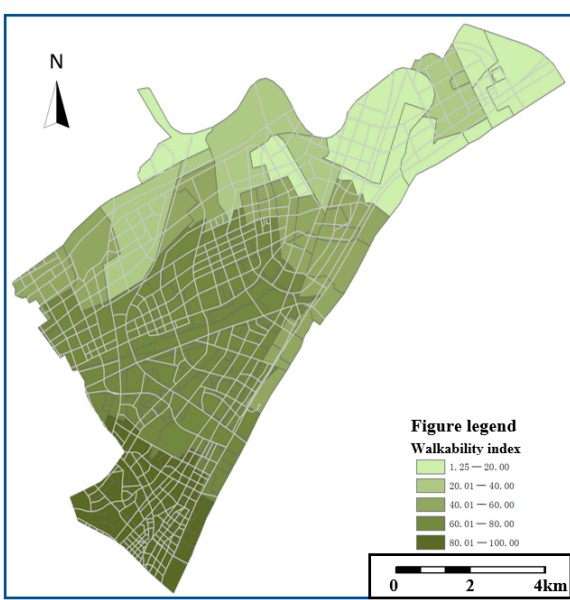

**Figure 11.** Walkability index of the communities in the Jiang'an district.

### 3.2.5. Income

This study used two indicators to characterize this factor: *population density (PopuDen)* (the number of residents per unit area) and income. Yet it was difficult to obtain income data due to privacy issues at the community level. Using web crawler, we obtained 5707 local housing prices from real estate websites which covered most of the new buildings in study area. The *average housing price (AHP)* was calculated as the income indicator instead.

### *3.3. Evaluating the Influences of Built Environment Indicators on the Prevalence of Hypertension*

### 3.3.1. Multivariate Regression Model

A mathematical model was used to analyze the correlation of the built environment and hypertension. SPSS software was used to incorporate the variables into the model using the entry methods in stepwise regression methods. After the regression equation was established, the significance of the dependent variable by the 95% confidence rule and the

magnitude of the effect were checked. The Moran's I index was used to analyze whether the distribution of residuals was normal or not.

### 3.3.2. Spatial Analysis Based on SLM and SEM

According to a community health survey conducted in 2008 in New York City, hypertension and diabetes prevalence exhibited spatial clustering similar to infectious diseases [52]. Thus, spatial econometric models are important [53,54] in such studies. A *spatial lag model (SLM)* and *spatial error model (SEM)* were established in the Geoda software. The difference in the degree of fitness and the level of interpretation between the spatial model and the linear model were compared to determine the association between the built environment and NCDs.

## 4. Results

### 4.1. Spatial Clustering of the Hypertension Prevalence

The hypertension prevalence data at the community level was imported into ArcGIS, and the *Kernel density analysis* tool was used. Spatial clustering of the hypertension prevalence was observed; some data clouds were concentrated at the lower left corner in the Jiang'an district, which is a downtown area with high density and high-rise buildings. Some areas showed a dotted distribution with others being blanks (Figure 12a).

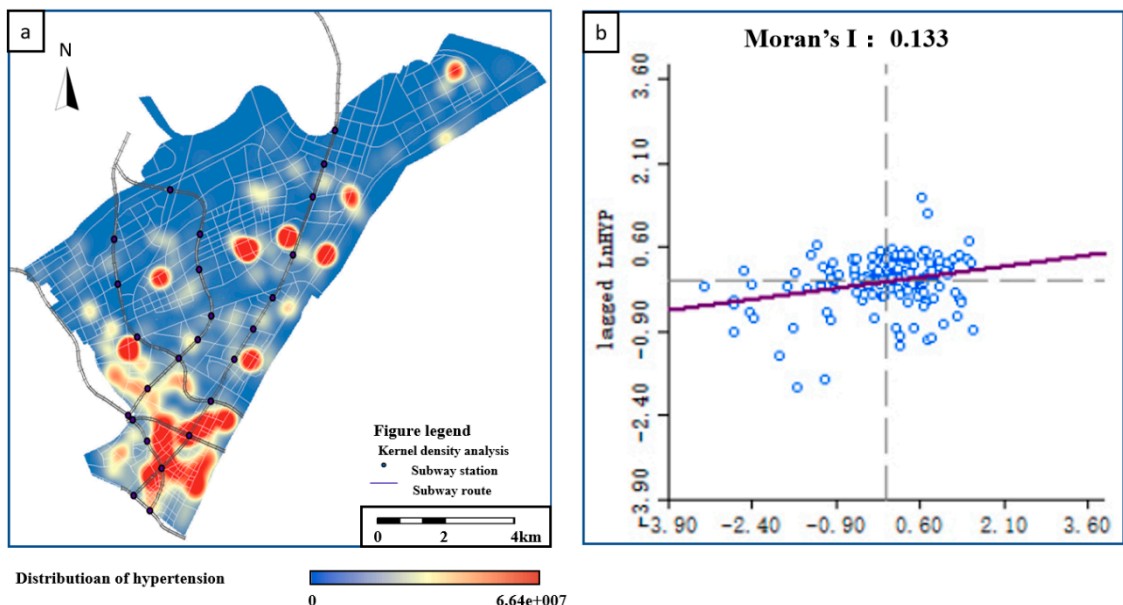

**Figure 12.** (**a**) Kernel density analysis of distribution of the hypertension patients; (**b**) scatter plot of Moran's I index.

Moran's I index was 0.133 (Z = 2.55, $p$ = 0.01), indicating spatial clustering of the hypertension prevalence. The scatter chart confirmed this result (Figure 12b). Hot spot analysis (*Getis-Ord Gi*) indicated 3 hot spots and 1 cold spot in the study area ($p < 0.05$). After performing cluster and outlier analysis (*Anselin Local Moran's I*), 2 high–high clustering communities, 4 low–low clustering communities, 3 low–high clustering communities, and 4 high–low clustering communities were observed in the study area (Figure 13).

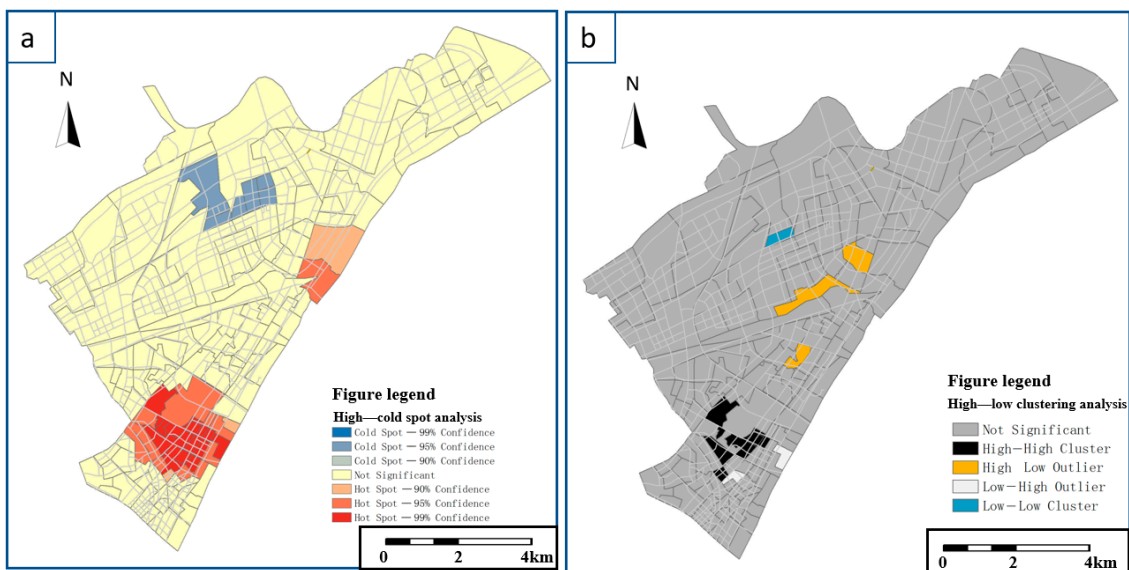

**Figure 13.** (**a**) Hot–cold spot analysis by Getis-Ord Giand and Anselin Local Moran's I; (**b**) results of high–low clustering of hypertension prevalence at the community level.

### 4.2. Results of Multivariate Regression Models

Based on the research framework (Figure 1), the *LUM, FAR, NDVI, MedCost, GymCost, AHP, WalkIndex, RoadDen, BusIndex*, and *PopuDen* were used as the explanatory variables. The prevalence of community hypertension was an independent variable. The results are shown in Table 1. The P-values of the models were all less than 0.01, indicating that the built environment factors significantly affect the prevalence of hypertension (confidence ratio > 99%). The all-variables model (model 1) had an $R^2$ of 0.517 and an *adjusted $R^2$* of 0.210. However, some variables in models 1 and 2 had relatively large variance inflation factor (VIF) values, indicating the existence of multicollinearity in the model. The optimal model (model 5) was obtained after excluding the covariant variables. The $R^2$ was 0.483, and the adjusted $R^2$ (0.198) was slightly lower than the all-variables model. The Durbin–Watson statistic was 2.058, indicating that no spacial clustering existed in residuals (Table 1). The scatter plot confirms this result (Figure 14).

**Table 1.** Model summary.

| Model | R | R-Square | Adjusted R-Square | Sig. | F | Std. Error of the Estimate | Durbin–Watson (U) |
|---|---|---|---|---|---|---|---|
| 1 | 0.517 | 0.267 | 0.210 | 0.000 | 4.696 | 37.867 | |
| 2 | 0.515 | 0.265 | 0.214 | 0.000 | 5.213 | 37.955 | |
| 3 | 0.507 | 0.257 | 0.212 | 0.000 | 5.678 | 38.353 | |
| 4 | 0.494 | 0.244 | 0.204 | 0.000 | 6.083 | 39.054 | |
| 5 | 0.483 | 0.233 | 0.198 | 0.000 | 6.729 | 39.624 | 2.058 |

[1] Predictors: (Constant), GymCost, LUM, PopuDen, Income, RoadDen, NDVI, Medcost, FAR, WalkIndex, BusIndex. [2] Predictors: (Constant), GymCost, PopuDen, Income, RoadDen, NDVI, Medcost, FAR, WalkIndex, BusIndex. [3] Predictors: (Constant), GymCost, PopuDen, Income, RoadDen, NDVI, Medcost, FAR, WalkIndex. [4] Predictors: (Constant), GymCost, Income, RoadDen, NDVI, Medcost, FAR, WalkIndex. [5] Predictors: (Constant), GymCost, Income, RoadDen, Medcost, FAR, WalkIndex.

Model 5 includes *GymCost, AHP, RoadDen, Medcost, FAR,* and *WalkIndex*. The statistics of the variables are shown in Table 2. The *p*-values of all variables were < 0.05, and the VIF values were < 7.5, indicating no significant collinearity between the variables (Table 2).

However, the *LUM, Busindex, NDVI, and PopuDen* did not pass the significance test. Hypertension is caused by many influencing factors besides genetic and behavioral factors,

while it is affected by complex system factors. Culture, socioeconomic status, and even food have significant influences.

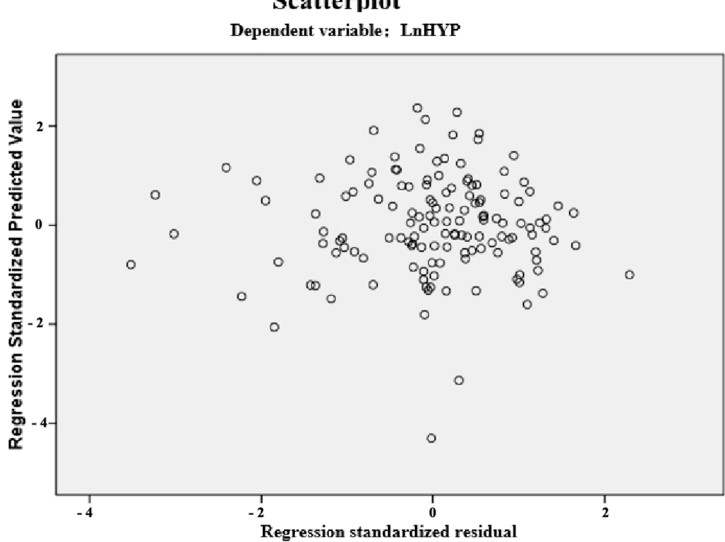

**Figure 14.** Scatter plot of model residuals.

**Table 2.** Model 5 variable statistics.

| Model | | Unstandardized Coefficients | | Standardized Coefficients | t | Sig. | Collinearity Statistics | |
|---|---|---|---|---|---|---|---|---|
| | | B | Std. Error | Beta | | | Tolerance | VIF |
| | (Constant) | −2.426 | 1.848 | | −1.312 | 0.192 | | |
| | Medcost | 0.496 | 0.177 | 0.325 | 2.796 | 0.006 | 0.428 | 2.335 |
| | Income | 1.028 | 0.396 | 0.235 | 2.593 | 0.011 | 0.704 | 1.421 |
| 5 | WalkIndex | −1.168 | 0.436 | −0.475 | −2.678 | 0.008 | 0.183 | 5.460 |
| | FAR | 0.253 | 0.152 | 0.213 | 1.660 | 0.099 | 0.349 | 2.865 |
| | RoadDen | −0.741 | 0.150 | −0.540 | −4.929 | 0.000 | 0.480 | 2.082 |
| | GymCost | 1.070 | 0.279 | 0.775 | 3.843 | 0.000 | 0.142 | 7.054 |

*4.3. The Performance of SEM and SLM*

Analysis showed that there was a more obvious spatial autocorrelation in the prevalence of hypertension at the community level. *Adjusted $R^2$* of the linear model was not high, indicating that the linear model was insufficient to explanation the spacial phenomena. Another characteristic of our hypertension prevalence data is that it carries spatial information. Therefore, spatial models were used to test the robustness of the model. The results are shown in the following table. Geoda software developed by Dr. Luc Anselin and his team was used to test the model by ordinary least squares (OLS), SEM, and SLM methods.

Comparison showed the fit superiority of the spatial model exceeded that of the linear model across the board. The results are shown in the table below (Table 3). The $R^2$ of the SLM model was higher (0.192) than that of the OLS (0.167) model. The *Akaike information criterion (AIC)* of the SEM model (279.003) was lower than that of the OLS model (281.505). The *Schwarz criterion* of the SEM model was also lower than that of the OLS model. Generally speaking, the lower these two indicators are, the better the fitting excellence of the model. Furthermore, the fitting performance of the SEM was slightly better than that of the SLM model.

**Table 3.** Comparison of the spatial lag model, spatial error model, and OLS model.

| | R-Squared | Adjusted R-Squared | AIC | Log-Likelihood | Schwarz Criterion |
|---|---|---|---|---|---|
| OLS regression model | 0.167 | 0.104 | 281.505 | −129.752 | 314.173 |
| Spatial lag model (SLM) | 0.192 | — | 280.400 | −128.200 | 316.038 |
| Spatial error model (SEM) | 0.190 | — | 279.003 | −128.501 | 311.670 |

The SEM model provided a better fit than the SLM model, indicating that the spatial distribution of NCDs was likely affected by the error in the spatial distribution and not by the influence of adjacent regions.

## 5. Discussion

### 5.1. The Significance of Influences of Built Environment Variables on Hypertension

The purpose of this study was not to establish an accurate regression model but to explore the potential association between the built environment factors and hypertension. The variables of model 1 were sorted by their significance and impact direction. The mean of the absolute value of the coefficient (0.629) was used as a threshold to evaluate the degree of impact (Table 4). The variables that had significant influence were *RoadDen, GymCost, Income, Medcost* and *WalkIndex*, while the ones having no significant influence were *NDVI, PopuDen, BusIndex,* and *LUM*.

**Table 4.** Summary of the variable correlations.

| Variables | Coefficients | Influence Direction and Degree | Sig. | Significance | Significance Ranking |
|---|---|---|---|---|---|
| RoadDen | −0.779 | − − | 0.000 | ** | 1 |
| GymCost | 1.401 | + + | 0.001 | ** | 2 |
| Income | 1.160 | + + | 0.005 | ** | 3 |
| Medcost | 0.510 | + | 0.006 | * | 4 |
| WalkIndex | −0.952 | − − | 0.042 | * | 5 |
| FAR | 0.281 | + | 0.071 | − | 6 |
| NDVI | 0.216 | + | 0.113 | Not significant | 7 |
| PopuDen | 0.024 | + | 0.132 | Not significant | 8 |
| BusIndex | −0.674 | − − | 0.251 | Not significant | 9 |
| LUM | 0.288 | + | 0.586 | Not significant | 10 |

** Indicates significance at the 0.01 level (paired *t*-test). * Indicates significance at the 0.05 level (paired *t*-test). + + Influence is positive and the coefficient is greater than 0.629; + influence is positive and the coefficient is lower than 0.629. − − Influence is negative and the absolute value of the coefficient is greater than 0.629; − influence is negative and the absolute value of the coefficient is lower than 0.629.

### 5.2. Built Environment Factors that Affect the Prevalence of Hypertension Significantly

Among the significant built environment factors, the *RoadDen* and *WI* had a negative influence on the prevalence of hypertension, which is consistent with most current studies. A high density of the road network indicates that the public service system in this area has good accessibility [33], resulting in a lower BMI index and lower risk of obesity [55,56]. It also indicates *a small community with a dense road network* layout block in which people are willing to go out walking and shopping because of commercial prosperity. The higher the *WI* of the street, the denser is the road network, which means it is more suitable for physical

activities, leading to a lower prevalence of hypertension. On the other hand, the cost of medical and sports facilities were positively correlated with the prevalence of hypertension. The less accessible the hospital facilities, the more difficult it is for residents to access health care. Similarly, we can conclude that the less accessible the gymnasium facilities, the more difficult it is for them to participate in sports, so the risk of NCDs is increased.

From the components of land use, *FAR* was positively correlated with the prevalence of hypertension, differing from the results of some recent studies. For example, Dunphy et al. found that physical activity, including walking and cycling, increased significantly when the population exceeded 7500 per square mile [57]. Many subsequent studies have confirmed this result [58]. However, our results are also consistent with those of some recent research. In a recent cross-sectional study, Sarkar et al. (2017) found a nonlinear relationship between the built environment factors and obesity [20]. Beside this, our result is also consistent with the findings of most studies in China. Sun et al. found that high-density environments increased the risk of obesity in Chinese cities [59]. The reason is that high-density development reduces urban public areas and green spaces, so that decreases the residents' willingness to be physically active [60].

*AHP (income indicator)* was positively correlated with the prevalence of hypertension. This is different from the usual perception [61]. High-income area residents have a better living environment and enough time for physical activity, so their prevalence of NCDs should be decreased. In the U.S., it was found that increases in income significantly improve mental and physical health but increase the prevalence of alcohol consumption [62]. There are three potential reasons to explain our results. First, the sample size in Jiang'an district was still so limited that it created statistical bias. Second, *AHP* is just an approximate estimate of income level. Price of real estate are determined by a number of factors, not only income factor, e.g., location, age of the housing, and even school district. Third, in Chinese cities, high-price areas are often located in the heart of the city with convenient public services. People's desire to go downstairs for physical activity and leisure is low. Another risk that cannot be ignored is that overeating and high alcohol consumption often increase the risk of CVD. This is regardless of whether a person is rich or not.

There is also an inverse relationship between the prevalence of hypertension and the distance cost of health care facilities. This is consistent with the findings of some previous studies. In the USA, inpatient utilization decreased when travel distance to VA facilities increased [63]. It is natural that if people are farther away from health care facilities, they will not have access to convenient health care services.

### 5.3. Other Built Environment Factors That Did Not Pass Significance Test

The *NDVI* ($p < 0.113$) and *PopDen* ($p < 0.132$) were not significant but we can see the effects of their impact. The *NDVI* was positively correlated with the prevalence of hypertension, which is contrary to some research results but consistent with others. Tilt et al. used objective measurements and self-assessment methods to study the influence of destination accessibility and *NDVI* on walking willingness and *BMI*. The data showed that residents living in areas with high green space accessibility and *NDVI* had lower *BMI* [64]. Yet, there were different results in some research. A study in the United Kingdom considered that there is not a clear link between the amount of green space and health outcomes [65]. Even more green space was associated with poor health status in low-income suburbs [66]. Another study in Netherland declared that people with more green space spend more time gardening and less time walking and cycling [67].

*Green space* is used differently in China than in the West. In China, due to the dense population, green space is very limited and generally green space is used as landscape and access is generally not allowed, nor is walking in it. In contrast, green spaces in the United States and Europe are easily accessible and used as fitness areas as a public recreational space. This is perhaps the main reason why Chinese green space indicators are different from those of the West.

Although the *NDVI* is still not a good indicator to measure green space in this research because wasteland and swamps are often considered green space, cloud and smog often interferes with the value of *NDVI*, so the *NDVI* is still an objective assessment of the green space relative to the green area ratio derived from the drawings. Further studies on indicators such as accessibility, distance to the nearest green space from home, visual/scenic quality of the views, and window views over the green space should be done in the future.

*PopDen* did not pass the significance test. A reason for this may be that the prevalence of hypertension has been standardized twice based on the population. However, the impact direction of *PopDen* is positive, supporting the conclusion that crowding is a risk factor for NCDs [68]. Research in a Chinese city found that a high-density environment reduces the residents' well-being. Noise and congestion may be the main reasons for residents' psychological stress [59].

The *BusIndex* was not significant either but its impact direction was negative. A study by Rundle et al. found that *BusIndex* significantly affected the BMI of New York's residents. Our research results support this conclusion.

The impact of *LUM* could not be confirmed ($p < 0.586$), which was not consistent with the findings of existing researches. In this study, the results showed that the *LUM* obtained from the entropy method was not suitable for the study area in China. The *LUM* obtained from land-use plans was quite different from actual conditions.

## 6. Conclusions

Urban studies differ from other sciences because of their objects and scale, and most studies do not allow the possibility of conducting multiple comparative experiments. Empirical case studies play an important role in advancing knowledge in the field of urban planning. Due to the lack of high-precision data in the past, most of our simulations of cities are too rough and abstract.

This study analyzed the correlation between the built environment factors and the prevalence of hypertension based on the 2015 Epidemiological Survey of people under 65 years in 144 communities of Wuhan city. While *RoadDen, GymCost, Income, Medcost* and *WalkIndex* had significant influence on the prevalence of hypertension, the correlation of *RoadDen, GymCost, Income, Medcost* and *WalkIndex,* the *NDVI, PopuDen, BusIndex,* and *LUM* are insignificant. In addition, it was found that the spatial models fit better than the multivariate linear model in the correlation analysis.

Our study is one of a few to systematically investigate the association of built environment factors with hypertension and quantify the dose–response relationship. In addition to a linear regression model, we also used the spatial econometric method to verify the robustness. These design features provided details and statistical rigor of the analysis. Despite the considerable complexity and uncertainties of healthy city planning and weak evidence of causality, decision-makers and researchers should not refuse to use the best available research evidence to plan and design healthy cities.

The inference that we can learn which built environment factors are detrimental to health and which factors can be improved through an improved living environment is a conclusion with significant public health implications. It needs to be carefully considered by urban planners and policy makers. Further longitudinal studies based on cumulative data are needed to clarify changes in the built environment and to infer causal relationships with health.

**Author Contributions:** Conceptualization, H.X.; Data curation, Y.Y.; Formal analysis, Y.M. and X.Z. (Xu Zhang); Funding acquisition, Q.W. and X.Z. (Xilin Zhou); Investigation, Y.Y.; Methodology, H.X.; Project administration, H.X.; Visualization, X.Z. (Xu Zhang); Writing – review & editing, X.Z. (Xilin Zhou). All authors have read and agreed to the published version of the manuscript.

**Funding:** This work was supported by the National Key Research and Development Program of China (Grant No. 2018YFC0704300); the Fundamental Research Funds for the Central Universities (Grant No. 2021IVA034); the Research Funds of Wuhan City Construction Bureau (Grant No. 201933).

**Data Availability Statement:** Not applicable.

**Acknowledgments:** Thanks are due to P. Zhang for assistance with the experiments and to X. Zhao for valuable discussion.

**Conflicts of Interest:** The authors declare no conflict of interest.

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
