# Peer review of "Built Environment Factors Influencing Prevalence of Hypertension at Community Level in China: The Case of Wuhan"

_sustainability, doi:10.3390/su13105580_

Round 1

Reviewer 1 Report

The manuscript presents an interesting, interdisciplinary research focused on the relationship between health and environmental (urban) context. Interdisciplinary character of work is its strong characteristic.

In terms of method applied, authors include abundance of aspects describing urban conditions. Probably grouping these factors and then correlating with health situation could give clearer findings.

In my opinion it is absolutely necessary to translate the data description in all of the images included in the text.  Please note that graphics provided constitute an integrate part of research and as such should be possible to interpret and assess.

Author Response

-Reviewer 1

The manuscript presents an interesting, interdisciplinary research focused on the relationship between health and environmental (urban) context. Interdisciplinary character of work is its strong characteristic.

In terms of method applied, authors include abundance of aspects describing urban conditions. Probably grouping these factors and then correlating with health situation could give clearer findings.

In my opinion it is absolutely necessary to translate the data description in all of the images included in the text. Please note that graphics provided constitute an integrate part of research and as such should be possible to interpret and assess.

-Responses:

  1. In terms of method applied, authors include abundance of aspects describing urban conditions. Probably grouping these factors and then correlating with health situation could give clearer findings.

Explanation:

Thanks for your detailed comments!

We did group the factors and correlated with the prevalence of hypertension in the Chapter 4.2. You can find those correlations in table 1. We are sorry to make you confused since the subtitle was not clear enough before. The structure of the whole manuscript had been re-organized. The revised contents related to your concerns are as below.

Revised contents:

Pg9, line 243-250

3.3. Evaluating influences of built environment indicators on the prevalence of hypertension

3.3.1 Multivariate regression model

Then mathematical model is used to analysis the correlation of built environment and hypertension. SPSS software was used to incorporate the variables into the model using the entry method. After the regression equation is established, the significance of the dependent variable by 95% confidence rule and the magnitude of the effect is checked. The Moran’s I index was used to analyze whether the distribution of residuals is normal or not.

Pg10, line 279-297

4.2. Results of multivariate regression models

Based on the research framework (Figure 1), the LUM, FAR, NDVI, MedCost, GymCost, AHP, WalkIndex, RoadDen, BusIndex, and PopuDen were used as the explanatory variables. The prevalence of community’s hypertension as independent variable. The results are shown in Table 1. The P-values of the models are all less than 0.01, indicating that the built environment factors significantly affect the prevalence of hypertension (confidence ratio>99%). The all-variables model (model 1) has an R2 of 0.517 and an adjusted R2 of 0.210. However, some variables in model 1 and 2 have relatively large variance inflation factor (VIF) values, indicating the existence of multicollinearity in the model. The optimal model (model 5) is obtained after excluding the covariate variables. The R2 is 0.483, and the adjusted R2 (0.198) is slightly lower than the all-variables model. Durbin-Watson statistic is 2.058, indicating that no spacial clustering existed in residuals (Table 1). The scatterplot confirms this result (Figure 13).

Table 1. Model Summary

Model

R

R-Square

Adjusted R-Square

Sig.

F

Std. Error of the Estimate

Durbin-Watson (U)

1

.517a

.267

.210

.000b

4.696

37.867

2

.515b

.265

.214

.000c

5.213

37.955

3

.507c

.257

.212

.000d

5.678

38.353

4

.494d

.244

.204

.000e

6.083

39.054

5

.483e

.233

.198

.000f

6.729

39.624

2.058

1 Predictors: (Constant), GymCost, LUM, PopuDen, Income, RoadDen, NDVI, Medcost, FAR, WalkIndex, BusIndex

2 Predictors: (Constant), GymCost, PopuDen, Income, RoadDen, NDVI, Medcost, FAR, WalkIndex, BusIndex

3 Predictors: (Constant), GymCost, PopuDen, Income, RoadDen, NDVI, Medcost, FAR, WalkIndex

4 Predictors: (Constant), GymCost, Income, RoadDen, NDVI, Medcost, FAR, WalkIndex

5 Predictors: (Constant), GymCost, Income, RoadDen, Medcost, FAR, WalkIndex

Pg12, Line 327-334:

5.1. The significances of influences of bulit environment variables on hypertension

The purpose of this study was not to establish an accurate regression model but to explore the potential association between the built environment factors and hypertension. The variables of model 1 are sorted by their significance and impact direction. The mean of the absolute value of the coefficient (0.629) was used as a threshold to evaluate the degree of impact (Table 4). While the variables which have significant influence on are RoadDen, GymCost, Income, Medcost and WalkIndex, the ones have no significant influences are NDVI, PopuDen, BusIndex and LUM.

  1. In my opinion it is absolutely necessary to translate the data description in all of the images included in the text. Please note that graphics provided constitute an integrate part of research and as such should be possible to interpret and assess.

Explanation:

Thanks for your kind consideration and suggestions!

Revised contents:

All the images associated with the data description were translated and re-mapped.

Reviewer 2 Report

General Comment

  1. The author needs to restructure the manuscript for being clear for reader. Results, discussion and conclusion sections need to be restructured.
  2. In the discussion section the author needs to discuss the significant results comparing with other study and add if possible the explanations or hypothesis to support your results
  3. In the conclusion section the author needs to display the main findings or the relevance of this research. The author doesn’t need to add or to quote one citation in the conclusion.

Specific Comment

4-Line 466: Individual income was correlated with hypertension. The author needs to discuss this result comparing with other studies. The same with the line 471 related to the medical cost correlated to hypertension

5-Table3: 444-452: The author need to shown the important of this table, and establish the relation with his topic. The reader needs to known the explanation of the line 446-447

6- Line 508. The NDVI was positively correlated with the prevalence of hypertension, which is contrary to some research but with others. The author needs to add the reference and discuss about.

7-Line 338: Methods: I think it’s better to use statistical analysis instead method 

Please consider my suggestions and comment. Thanks so much

Author Response

Reponses to Reviewer 2

-Reviewer 2

  • General Comment
  1. The author needs to restructure the manuscript for being clear for reader. Results, discussion and conclusion sections need to be restructured.
  2. In the discussion section the author needs to discuss the significant results comparing with other study and add if possible the explanations or hypothesis to support your results
  3. In the conclusion section the author needs to display the main findings or the relevance of this research. The author doesn’t need to add or to quote one citation in the conclusion.
  • Specific Comment

4-Line 466: Individual income was correlated with hypertension. The author needs to discuss this result comparing with other studies. The same with the line 471 related to the medical cost correlated to hypertension

5-Table3: 444-452: The author need to show the important of this table, and establish the relation with his topic. The reader needs to know the explanation of the line 446-447

6- Line 508. The NDVI was positively correlated with the prevalence of hypertension, which is contrary to some research but with others. The author needs to add the reference and discuss about.

7-Line 338: Methods: I think it’s better to use statistical analysis instead method

Please consider my suggestions and comment. Thanks so much

-Responses:

  1. The author needs to restructure the manuscript for being clear for reader. Results, discussion and conclusion sections need to be restructured.

Responses:

Thanks for your kind comments and summary!

Revised contents:

The structure of the manuscript has been restructured to be easier for reading. The revised contents (subtitles, sentence or words) were colored into red throughout the manuscript.

  1. In the discussion section the author needs to discuss the significant results comparing with other study and add if possible the explanations or hypothesis to support your results

Responses:

Thanks for your kind comments and summary!

After the re-organization of the manuscript, the discussion section is easier to read now. The literatures which we compared with were colored into blue so as to help readers to catch the literatures.

Revised contents:

Pg12-13, Line 337-361:

5.2 The significant built environment factors on the prevalence hypertension

Among the significant built environment factors, the RoadDen and WI had a negative influence on the prevalence of hypertension, which is consistent with most current studies. A high density of the road network indicates that the public service system in this area has good accessibility (Frank et al., 2004), resulting in a lower BMI index and lower risk of obesity (Oluyomi, 2011; Ball et al., 2012). It also means of a small community with a dense road network layout block in which people are willing to go out walking and shopping because of commercial prosperity. The higher the WI of the street, the denser the road network, means the more suitable for physical activities, leading to lower prevalence of hypertension. On the other side, the cost of medical and sports facilities were positively correlated with the prevalence of hypertension. The less accessible the hospital facility, the more difficult it is for them to access health care. Similarly, we can conclude that the less accessible the gymnasiums facility, the more difficult it is for them to participate in sports, so the risk of NCDs is increased.

From the components of land use, FAR was positively correlated with the prevalence of hypertension, differing from the results of some recent studies. For example, Dunphy et al. found that physical activity, including walking and cycling, increased significantly when the population exceeded 7,500 per square mile (Dunphy and Fisher, 1996). Many subsequent studies have confirmed this result (Li et al., 2009; Rundle et al., 2007a). However, our results are also consistent with those of some recent research. In a recent cross-sectional study, Sarkar et al. (2017) found a non-linear relationship between the built environment factors and obesity (Sarkar et al., 2017). Beside this, our result is also consistent with the findings of most studies in China. Sun et al. found that high density increased the risk of obesity in Chinese cities (Sun et al., 2017). The reason is that high density reduces urban public areas and green spaces, so that decrease the residents’ willing of physical activity (Alfonzo et al., 2014).

  1. In the conclusion section the author needs to display the main findings or the relevance of this research. The author doesn’t need to add or to quote one citation in the conclusion.

Responses:

Thanks for your kind consideration and suggestions!

Revised contents:

Pg 14, Line 420-445

6.Conclusions

Urban study differs from other sciences because of its object and scale of study, most of which do not have the possibility of conducting multiple comparative experiments. Empirical case studies play an important role in advancing knowledge in the field of urban planning. Due to the lack of high precision data in the past, most of our simulations of cities are too rough and abstract.

This study analyzed the correlation between the built environment factors and the prevalence of hypertension based on the 2015 Epidemiological Survey of people under 65 years in 144 communities of Wuhan city. While the RoadDen, GymCost, Income, Medcost and WalkIndex have significant influence on prevalence of hypertension, the correlation of RoadDen, GymCost, Income, Medcost and WalkIndex, the NDVI, PopuDen, BusIndex and LUM are insignificant. In addition, it was found that the spatial models fit better than the multivariate linear model in the correlation analysis.

Our study is one of a few to systematically investigate the association of built environment factors with hypertension and quantify the dose-response relationship. In addition to a linear regression model, we also used the spatial econometric method to verify the robustness. These design features provided details and statistical rigor of the analysis. Despite the considerable complexity and uncertainties of healthy city planning and weak evidence of causality, decision-makers and researchers should not refuse to use the best available research evidence to plan and design healthy cities.

The inference that we can learn which built environment factors are detrimental to health and which factors can be improved through improved living environment is a conclusion with significant public health implications. It needs to be carefully considered by urban planners and policy makers. Further longitudinal studies based on cumulative data are needed to clarify changes in the built environment and to infer causal relationships with health.

  1. Line 466: Individual income was correlated with hypertension. The author needs to discuss this result comparing with other studies. The same with the line 471 related to the medical cost correlated to hypertension.

Responses:

Thanks for your detailed comments!

Two literatures were added to discuss the income and health.

Revised contents:

Pg 13, Line 361-374

AHP (Income indicator) was positively correlated with the prevalence of hypertension. This is different from the usual perception (Michael Marmot, 2002). High-income areas have a better living environment and enough time for physical activity, their prevalence of NCDs should be decreased. In the U.S., it was found that increases in income significantly improve mental and physical health but increase the prevalence of alcohol consumption. (Susan L. Ettner,1996). There are three potential reasons to explain our results. First, the sample size in Jiang’an district is still so limited that creates statistical bias. Second, AHP is only an approximate of income level since real estate prices depend on more than only income, it also affected by other factors, such as location, age of the house even school district or not. The Third, in Chinese cities, high-price areas often located in the heart of the city with convenient public services. People's desire to go downstairs for physical activity and leisure is low. Another risk that cannot be ignored is that over-eating food and high alcohol consumption often increases the risk of CVD. This is regardless of whether rich or not.

There is also an inverse relationship between the prevalence of hypertension and the distance cost of health care facilities. This is consistent with the findings of some previous studies. In the USA, inpatient utilization decreased when travel distance to VA facility increased(LaVela et al., 2004). It is natural if people are farther away from health care facilities, they will not have access to convenient health care services.

  • Michael Marmot. (2002) The influence of income on health: views of an epidemiologist. Health Affairs, 21(2),31-46. DOI: 10.1377/hlthaff.21.2.31
  • Susan L. Ettner. (1996) New evidence on the relationship between income and health. Journal of Health Economics, 15(01), 67-85. Available at: https://doi.org/10.1016/0167-6296(95)00032-1
  • LaVela S.L, Smith B. and Weaver F.M., et al. (2004) Geographical proximity and health care utilization in veterans with SCI&D in the Social Science and Medicine 59: 2387-2399. https://doi.org/10.1016/j.socscimed.2004.06.033
  1. Table3: 444-452: The author need to show the important of this table, and establish the relation with his topic. The reader needs to know the explanation of the line 446-447

Responses:

Thanks for your detailed comments!

Revised contents:

Pg 11, Line 301-321

4.3. The performances of SEM and SLM

After analysis, it is known that there is a more obvious spatial autocorrelation in the prevalence of hypertension in the community level. Adjusted R2 of the linear model was not high, indicating that the linear model is insufficient to explanation the spacial phenomena. Another characteristic of our hypertension prevalence data is that it carries spatial information. Therefore, spatial models is used to test the robustness of the model. The results are shown in the following table. Geoda software developed by Dr. Luc Anselin and his team was used to test by ordinary least squares (OLS) model, SEM, and SLM.

After comparison, the fitting excellence of the spatial model exceeds that of the linear model across the board. The results are shown in the table below (Table 3). The R2 of SLM model was higher (0.192) than the OLS (0.167) model. The Akaike information criterion (AIC) of SEM model (279.003) was lower than the OLS model (281.505). The Schwarz criterion of SEM model was also lower than the OLS model. Generally speaking, the lower these two indicators are, the better the fitting excellence of model. Furthermore, the fitting performance of the SEM was slightly better than that of the SLM model.

Table3. Comparison of the spatial lag model, spatial error model, and OLS model

R-squared

Adjusted R-squared

AIC

Log-likelihood 

Schwarz criterion    

OLS regression model

0.167

0.104

281.505

-129.752

314.173

Spatial lag model

0.192

280.400

-128.200

316.038

Spatial error model

0.190

279.003

-128.501

311.670

The SEM model provided a better fit than the SLM model, indicating that the spatial distribution of NCDs was likely affected by the error in the spatial distribution and not by the influence of adjacent regions.

  1. Line 508. The NDVI was positively correlated with the prevalence of hypertension, which is contrary to some research but with others. The author needs to add the reference and discuss about.

Responses:

Thanks for your detailed comments!

Revised contents:

Pg2, Line 71-72

In addition, it was reported that green space is beneficial for mental health [21,22], not only of designated green space types such as the park but also, and in general, street greenery [23].

References:

  • Beute, F., Andreucci, M.B., Lammel, A., Davies, Z., Glanville, J., Keune, H., Marselle, M., O’Brien, L.A., Olszewska-Guizzo, A., Remmen, R., Russo, A., & de Vries, S. (2020) Types and characteristics of urban and peri-urban green spaces having an impact on human mental health and wellbeing.Report prepared by an EKLIPSE Expert Working Group.UK Centre for Ecology & Hydrology, Wallingford, United Kingdom. Available at: https://eklipse.eu/wp-content/uploads/website_db/Request/Mental_Health/EKLIPSE_HealthReport-Green_Final-v2-Digital.pdf
  • Olszewska-Guizzo, A., Sia, A., Fogel, A., & Ho, R. (2020). Can exposure to certain urban green spaces trigger frontal alpha asymmetry in the brain? — Preliminary findings from a passive task EEG study. International journal of environmental research and public health, 17(2), 394. https://doi.org/10.3390/ijerph17020394
  • Andreucci, M. B., Russo, A., & Olszewska-Guizzo, A. (2019). Designing urban green-blue infrastructure for mental health and elderly wellbeing. Sustainability, 11(22), 6425. https://doi.org/10.3390/su11226425

Pg 13, Line 394-399

5.3 The insignificant built environment factors on the prevalence hypertension

Green space is used differently in China than in the West. In China, due to the dense population, green space is very limited and generally green space is used as a landscape and access is generally not allowed, nor is walking in it. In contrast, green spaces in the United States and Europe are easily accessible and used as fitness areas as a public recreational space. This is perhaps the main reason why Chinese green space indicators are different from those of the West.

Although NDVI is still not a good indicator to measure green space in this research because wasteland and swamps often considered green space, cloud and smog often interferes with the value of NDVI, the NDVI is still an objective assessment of the green space relative to the green area ratio derived from the drawings. Further studies on the indicators such as accessibility, distance to the nearest green space from home, visual/ scenic quality of the views, window views over the green space should be done in the future.

  1. Line 338: Methods: I think it’s better to use statistical analysis instead method

Responses:

Thanks for your detailed comments!

The subtitles were revised.

Revised contents:

Pg 9, Line 240-255

3.3. Evaluating the influences of built environment indicators on the prevalence of hypertension

3.3.1 Multivariate regression model

Then mathematical model is used to analysis the correlation of built environment and hypertension. SPSS software was used to incorporate the variables into the model using the entry method. After the regression equation is established, the significance of the dependent variable by 95% confidence rule and the magnitude of the effect is checked. The Moran’s I index was used to analyze whether the distribution of residuals is normal or not.

3.3.2 Spatial analysis based on SLM and SEM

According to a community health survey conducted in 2008 in New York City, hypertension and diabetes prevalence exhibited spatial clustering similar to infectious diseases (Hygiene department of New York, 2008). Thus, spatial econometric models are necessary (Anselin, 1988; Anselin and Florax, 1995). The Spatial lag model (SLM) and the Spatial error model (SEM) were established in the Geoda software. The difference in the degree of fitness and the level of interpretation between the spatial model and the linear model were compared to determine the association between the built environment and NCDs.

Reviewer 3 Report

This is an interesting study shedding light on the environmental determinants of cardiovascular health in a large Chinese city. 

Interesting Introduction but the section about the green spaces and psychological health could be extended as there is a lot of new and interesting research emerging nowadays, and new interesting mechanisms are being discovered, broader than ones from 70-80. which Authors cite. Some possible recent literature to include:

1) "Types and characteristics of urban and peri-urban green spaces having an impact on human mental health and wellbeing: a systematic review"

2) Olszewska-Guizzo, A., Sia, A., Fogel, A., & Ho, R. (2020). Can exposure to certain urban green spaces trigger frontal alpha asymmetry in the brain?—Preliminary findings from a passive task EEG study. International journal of environmental research and public health17(2), 394.

3) Andreucci, M. B., Russo, A., & Olszewska-Guizzo, A. (2019). Designing urban green-blue infrastructure for mental health and elderly wellbeing. Sustainability11(22), 6425.

The study is conducted well using appropriate methods and sample. However, Authors should discuss the limitations encountered in the Discussion section.

Some more detailed comments:

[line 18] - Authors report the correlation of built factors and hypertension, so it is incorrect to say that the built environment has a sig.effect because it suggests causation.

[line 101] - LUM abbreviation should be explained the first time it appears in text.

[line 289-293] - the selection of the NDVI as only one indicator regarding the green space should be better justified in text. NDVI is usually used to estimate the biodiversity, and level of biodiversity may not directly translate to human perception of space. There are existing other indices such as accessibility, distance to the nearest green space from home, visual/ scenic quality of the views, window views over the green space (green space visual exposure, a contemplative score of the neighbourhood landscape). Later in the Discussion Authors could consider describing that as a limitation which could be linked with the lack of significant correlation of green spaces and hypertension. 

[line 294 - 296] - urban design can be measured and described using multiple technical and operationalized techniques. For example using the imageability index derived from GIS, visual openings rate (3D models), as well as traditional components of the city fabric as per Kevin Lynch theory ("Image of the City"). The authors choice seem modest and at the very least it is not true that "Urban design involves emotional and subjective feelings, making it difficult to obtain  a quantitative measure", so this should be explored better.

Author Response

Reponses to Reviewer 3

-Reviewer 3

This is an interesting study shedding light on the environmental determinants of cardiovascular health in a large Chinese city. Interesting Introduction but the section about the green spaces and psychological health could be extended as there is a lot of new and interesting research emerging nowadays, and new interesting mechanisms are being discovered, broader than ones from 70-80. which Authors cite. Some possible recent literature to include:

1) "Types and characteristics of urban and peri-urban green spaces having an impact on human mental health and wellbeing: a systematic review"

2) Olszewska-Guizzo, A., Sia, A., Fogel, A., & Ho, R. (2020). Can exposure to certain urban green spaces trigger frontal alpha asymmetry in the brain? — Preliminary findings from a passive task EEG study. International journal of environmental research and public health, 17(2), 394.

3) Andreucci, M. B., Russo, A., & Olszewska-Guizzo, A. (2019). Designing urban green-blue infrastructure for mental health and elderly wellbeing. Sustainability, 11(22), 6425.

Responses:

Thanks for your kind suggestions and summary!

The suggested literatures of green spaces and psychological health were added.

Revised contents:

Pg 2, Line 68-73

Sarkar et al. found that in a low-residential-density(1800p/km²) or less, a positive association between density and obesity was observed. If density over 1800/km², it was negatively associated with obesity on the contrast (Sarkar et al., 2017). In addition, it was reported that green space is beneficial for mental health (Andreucci, et al., 2019; Olszewska-Guizzo, et al., 2020), not only of designated green space types such as the park but also, and in general, street greenery. (Beute et al., 2020)

References:

Beute, F., Andreucci, M.B., Lammel, A., Davies, Z., Glanville, J., Keune, H., Marselle, M., O’Brien, L.A., Olszewska-Guizzo, A., Remmen, R., Russo, A., & de Vries, S. (2020) Types and characteristics of urban and peri-urban green spaces having an impact on human mental health and wellbeing. Report prepared by an EKLIPSE Expert Working Group.UK Centre for Ecology & Hydrology, Wallingford, United Kingdom. Available at: https://eklipse.eu/wp-content/uploads/website_db/Request/Mental_Health/EKLIPSE_HealthReport-Green_Final-v2-Digital.pdf

Olszewska-Guizzo, A., Sia, A., Fogel, A., & Ho, R. (2020). Can exposure to certain urban green spaces trigger frontal alpha asymmetry in the brain? — Preliminary findings from a passive task EEG study. International journal of environmental research and public health, 17(2), 394. https://doi.org/10.3390/ijerph17020394

Andreucci, M. B., Russo, A., & Olszewska-Guizzo, A. (2019). Designing urban green-blue infrastructure for mental health and elderly wellbeing. Sustainability, 11(22), 6425. https://doi.org/10.3390/su11226425

  • Some more detailed comments:
  1. [line 18] - Authors report the correlation of built factors and hypertension, so it is incorrect to say that the built environment has a sig. effect because it suggests causation.
  2. [line 101] - LUM abbreviation should be explained the first time it appears in text.
  3. [line 289-293] - the selection of the NDVI as only one indicator regarding the green space should be better justified in text. NDVI is usually used to estimate the biodiversity, and level of biodiversity may not directly translate to human perception of space. There are existing other indices such as accessibility, distance to the nearest green space from home, visual/ scenic quality of the views, window views over the green space (green space visual exposure, a contemplative score of the neighbourhood landscape). Later in the Discussion Authors could consider describing that as a limitation which could be linked with the lack of significant correlation of green spaces and hypertension.
  4. [line 294 - 296] - urban design can be measured and described using multiple technical and operationalized techniques. For example using the imageability index derived from GIS, visual openings rate (3D models), as well as traditional components of the city fabric as per Kevin Lynch theory ("Image of the City"). The authors choice seem modest and at the very least it is not true that "Urban design involves emotional and subjective feelings, making it difficult to obtain a quantitative measure", so this should be explored better.

-Responses:

  1. [line 18] - Authors report the correlation of built factors and hypertension, so it is incorrect to say that the built environment has a sig. effect because it suggests causation.

Responses:

Thanks for your kind comments!

The significant effect was replaced to “significant correlations”

Revised contents:

Pg1, Line 18

Result indicated built environment factors have significant correlations on the prevalence of hypertension.

  1. [line 101] - LUM abbreviation should be explained the first time it appears in text.

Responses:

Thanks for your kind comments!

Revised contents:

Pg1, Line 98-100:

There has been a large body of work on this (Takemi et al., 2014; N. N. et al., 2015; Ewing et al., 2014). Frank et al. (2004) found that every 25% increase in the land use mix(LUM) reduced the overweight and obesity rate by 12.2%.

  1. [line 289-293] - the selection of the NDVI as only one indicator regarding the green space should be better justified in text. NDVI is usually used to estimate the biodiversity, and level of biodiversity may not directly translate to human perception of space. There are existing other indices such as accessibility, distance to the nearest green space from home, visual/ scenic quality of the views, window views over the green space (green space visual exposure, a contemplative score of the neighbourhood landscape). Later in the Discussion Authors could consider describing that as a limitation which could be linked with the lack of significant correlation of green spaces and hypertension.

Responses:

Thanks for your kind consideration and suggestions!

The limitations of the NDVI were discussed.

Revised contents:

Pg 13, Line 400-406

Although NDVI is still not a good indicator to measure green space in this research because wasteland and swamps often considered green space, cloud and smog often interferes with the values and NDVI cannot directly translate to human perception of space, the NDVI is still an objective assessment of the green space relative to the green area ratio derived from the drawings. Further studies on the indicators such as accessibility, distance to the nearest green space from home, visual/ scenic quality of the views, window views over the green space should be done in the future.

  1. [line 294 - 296] - urban design can be measured and described using multiple technical and operationalized techniques. For example using the imageability index derived from GIS, visual openings rate (3D models), as well as traditional components of the city fabric as per Kevin Lynch theory ("Image of the City"). The authors choice seem modest and at the very least it is not true that "Urban design involves emotional and subjective feelings, making it difficult to obtain a quantitative measure", so this should be explored better.

Responses:

Thanks for your kind comments!

We make a further explanation in the section 3.2.4

Revised contents:

Pg 8, Line 211-224

3.2.4. Urban design

The morphological parameters such as sky view factor(SVF), street width to building height ratio (W/H), building coverage ratio (BCR) and Building surface fraction (BSF), etc. are key factors of urban design. While the morphological parameters were linked with the vibrancy of the city, they are too many to be examined in this study. Besides, the emotionally and subjective feelings of residents on the urban design make it difficult to obtain a quantitative measure (Ewing and Handy, 2009). The walkability index (WI) is a recently emerged indicator that evaluates the vibrancy of the city (Herrmann et al., 2017), indirectly reflecting the quality of urban design (Zhou et al., 2019; Carroll et al., 2016; Long and Zhou, 2016). Many algorithms exist to calculate the WI. This study used the walk score algorithm, which is relatively mature and is recognized by most scholars worldwide (NRDC, 2017). The walk score was calculated the usage of various public facilities with different weights based on the travel behaviors of pedestrians. The walking distance attenuation, intersection density, and road length were considered to improve the accuracy of the WI (Eq. 4) (Figure 10).

Round 2

Reviewer 3 Report

The authors have significantly improved the manuscript and addressed all my previous comments. I would recommend this work for publication.